# Characterisation of the Antibody Response in Sinopharm (BBIBP-CorV) Recipients and COVID-19 Convalescent Sera from the Republic of Moldova

**DOI:** 10.3390/vaccines11030637

**Published:** 2023-03-13

**Authors:** Mariana Ulinici, Alen Suljič, Monica Poggianella, Rafaela Milan Bonotto, Katarina Resman Rus, Angela Paraschiv, Amedeo Marco Bonetti, Mihail Todiras, Alexandru Corlateanu, Stanislav Groppa, Emil Ceban, Miroslav Petrovec, Alessandro Marcello

**Affiliations:** 1National Institute for Health and Medical Research, Faculty of Medicine, Nicolae Testemitanu State University of Medicine and Pharmacy, 2004 Chisinau, Moldova; 2Alfa Diagnostica Laboratory, 2021 Chisinau, Moldova; 3Institute of Microbiology and Immunology, Faculty of Medicine, University of Ljubljana, SI-1000 Ljubljana, Slovenia; 4Laboratory of Molecular Virology, International Centre for Genetic Engineering and Biotechnology (ICGEB), 34149 Trieste, Italy

**Keywords:** SARS-CoV-2, vaccine, Sinopharm, BBIBP-CorV, RBD ELISA, neutralisation assay, pseudovirus, convalescent plasma

## Abstract

The early availability of effective vaccines against SARS-CoV-2, the aetiologic cause of COVID-19, has been at the cornerstone of the global recovery from the pandemic. This study aimed to assess the antispike RBD IgG antibody titres and neutralisation potential of COVID-19 convalescent plasma and the sera of Moldovan adults vaccinated with the Sinopharm BBIBP-CorV vaccine. An IgG ELISA with recombinant SARS-CoV-2 spike RBD and two pseudovirus-based neutralisation assays have been developed to evaluate neutralising antibodies against SARS-CoV-2 in biosafety level 2 containment facilities. A significant moderate correlation was observed between IgG titres and the overall neutralising levels for each neutralisation assay (ρ = 0.64, *p* < 0.001; ρ = 0.52, *p* < 0.001). A separate analysis of convalescent and vaccinated individuals showed a higher correlation of neutralising and IgG titres in convalescent individuals (ρ = 0.68, *p* < 0.001, ρ = 0.45, *p* < 0.001) compared with vaccinated individuals (ρ = 0.58, *p* < 0.001; ρ = 0.53, *p* < 0.001). It can be concluded that individuals who recovered from infection developed higher levels of antispike RBD IgG antibodies. In comparison, the Sinopharm-vaccinated individuals produced higher levels of neutralising antibodies than convalescent plasma.

## 1. Introduction

The ongoing pandemic of the coronavirus disease 2019 (COVID-19), caused by severe acute respiratory syndrome coronavirus 2 (SARS-CoV-2), has claimed over six million lives worldwide and profoundly affected our society [1,2]. In the Republic of Moldova, the first case of SARS-CoV-2 infection was registered on 7 March 2020 [3], and, as of 27 November 2022, 595,073 COVID-19 cases (15,955 per 100,000 inhabitants) have been registered, causing 11,918 deaths (319.6 deaths/100,000 inhabitants).

Virus-specific vaccines and antiviral drugs are two common strategies to combat viral diseases [4]. Human neutralising antibodies (NAbs) targeting the host ACE2 receptor-binding domain (RBD) of the SARS-CoV-2 spike protein [5,6] have demonstrated therapeutic potential and are being tested in clinical trials [5,7,8]. These antibodies can be induced by vaccination or transferred as therapeutics in convalescent-phase sera [9].

In March 2021, 2000 doses of the Sinopharm vaccine were donated to the Republic of Moldova and were solely administered to students and professors of the Nicolae Testemitanu University [10]. The Sinopharm/BBIBP-CorV is an inactivated COVID-19 vaccine that received an emergency user licence from the WHO on 7 May 2021, to be supplied through the Vaccines Global Access (COVAX) programme [11], and it has been approved in more than 70 countries [12], including the Republic of Moldova.

Even with current immunisation campaigns to combat SARS-CoV-2, there remains a considerable need for viable therapeutic solutions [13]. Studies have demonstrated that NAbs against SARS-CoV-2 protect animal models from infection [14,15] and are being evaluated for prophylaxis and as therapeutics in humans [5].

A robust serological test to detect NAbs against SARS-CoV-2 was urgently needed in the country to determine the predicted humoral protection and vaccine efficacy after vaccination. According to the first interim national guideline on clinical management of patients with COVID-19 infection, convalescent plasma (CP) has been recognised as a potential treatment for critically ill COVID-19 patients [16]. According to this document, the infusions of freshly frozen plasma from recovered patients containing NAbs are administered to patients with severe or critical forms of COVID-19. The US Food and Drug Administration (FDA) has recommended that convalescent plasma with a virus-neutralising (VN) antibody titre of ≥1:160 be used for therapeutic transfusion [17]. Without quantitative assays to measure antibody titres and neutralising ability, the potential activity of donor plasma remains unknown prior to transfusion. Thus, accurate determinations of neutralising antibody titres are critical for surveillance and testing patient sera [18].

Neutralisation assays measure how effectively donor plasma or sera can inhibit virus infection and are the gold standard for measuring the antiviral activity of antibodies [18,19,20]. In the case of SARS-CoV-2, such assays require biosafety level 3 (BSL-3) containment facilities and highly trained personnel. To overcome this limitation, pseudotyped viruses have been developed as alternatives to infectious viruses, which can be handled at BSL-2 [21].

This is the first study in the country aiming to evaluate the levels of SARS-CoV-2 anti-RBD spike (S) and NAbs in 100 recovered patients and 100 individuals after a double dose of Sinopharm vaccine. To achieve this aim, an IgG ELISA with recombinant SARS-CoV-2 spike RBD and a protocol to produce pseudotyped lentiviruses expressing on the membrane the spike glycoprotein of SARS-CoV-2 D614G was developed. The latter was tested for SARS-CoV-2 neutralisation using either a cytofluorimeter or a high-content image analysis instrument on many sera from vaccinated individuals and infected individuals. This study highlights a wide phenotypic variation in human antibody responses against SARS-CoV-2. It demonstrates the efficacy of the lentivirus pseudotyped assay for high-throughput serological surveys of neutralising antibody titres in large cohorts [22].

## 2. Materials and Methods

### 2.1. Specimen Collection

In this study, the following samples were tested: (i) serum samples from vaccinated subjects, taken 14 days after the second dose of the Sinopharm COVID-19 vaccine (*n* = 100); (ii) convalescent plasma collected from patients with a negative result for COVID-19 from a PCR test, taken14 days after clinical recovery (*n* = 100); and (iii) 96 negative control samples collected in 2018. The vaccinated group intramuscularly received 2 doses (0.5 mL) of the Sinopharm vaccine. The vaccine contained 0.225 mg of aluminium hydroxide as adjuvant. Specimens were treated for 1 h at 56 °C to inactivate complement components and SARS-CoV-2 and stored at −80 °C until use. All plasma samples of COVID-19 patients and negative donors were obtained from the National Transfusion Centre, Republic of Moldova; the sera from vaccinated individuals were collected at the State University of Medicine and Pharmacy in collaboration with the ALFA Diagnostica laboratory, Republic of Moldova. The study and data collection were conducted with the approval of the Research Ethics Committee of Nicolae Testemitanu State University of Medicine and Pharmacy (Protocol No 3/24.01.22). All experiments were performed following relevant guidelines and regulations. Written informed consent was obtained from all participants. All patient data were anonymised before study inclusion.

### 2.2. Cell Lines

Human embryonic kidney 293 cells containing the SV40 T antigen (HEK293T, ATCC, Rockville, MD, USA, CRL-3216) were cultured in Dulbecco’s modified eagle’s medium (DMEM, with D-glucose (1 g/lit) and pyruvate (GIBCO Ref. 21885-025)); supplemented with 10% heat-inactivated foetal bovine serum (FBS) (GIBCO Ref. 10270-106) and 50 µg/mL gentamicin; and maintained at 37 °C, 5% CO_2_ and 80% humidity. Huh7 or HEK293 cells expressing the ACE2 receptor (Huh7-hACE2 and HEK293/ACE2) were produced through the transduction of a lentivector expressing human ACE2, as described previously [23]. Cells were grown in DMEM, supplemented with puromycin 1 µg/mL (Invitrogen) and maintained in an incubator at 37 °C and 5% of CO_2_. ACE2 expression level on the HEK293-ACE2 cell line was verified by FC. Briefly, 5 × 10^5^ cells were incubated with blocking solution (phosphate-buffered saline (PBS)—3%; bovine serum albumin (BSA)—0.2% NaN3) for 40 min at 4 °C and then incubated, in two reactions, for 1 h at 4 °C each with the following antibodies: primary antibody polyclonal goat IgG anti-ACE2 (1:50 dilution in blocking solution, antibody catalogue number AF933; Biotechne), secondary antibody fluorescein (FITC)-conjugated AffiniPure Rabbit antigoat IgG, FC γ-specific (1:500 dilution in blocking solution, code number 305-095-008; Jackson Immuno Research). After washing with PBS (0.2% NaN3), cells were resuspended in PBS (10 mM EDTA, pH 8), and the cytofluorimetry data were analysed using FlowJo software V10 (BD).

### 2.3. Pseudovirus Preparation

A lentivector system was used for the preparation of SARS-CoV-2 spike pseudoviruses expressing green fluorescent protein (GFP) [24,25]. HEK-293T cells were transfected with 3 plasmids: (i) pLVTHM, a coding for a GFP reporter; (ii) the packaging plasmid psPAX2; and (iii) pCDNA3, carrying the Wuhan-hu-1 D614G SPIKEΔCyto (gift from Prof. Massimo Pizzato, University of Trento).

HEK293T cells were seeded overnight and transfected by the calcium phosphate method with the 3 plasmids outlined above in the molar proportions 3:1.5:1, according to standard protocols [26]. After 5 days, the medium was collected, filtered through a 0.45 μM Millex-HV syringe filter, aliquoted and stored at −80 °C until use.

To assess the pseudovirus titre, 300.000 HEK293-ACE2 cells/well were seeded onto a 96-well plate (FALCON, Ref#353072) and transduced the next day with twofold dilutions of the pseudovirus in 2% FBS DMEM. After 3 days, cells were collected and resuspended in 5 mM EDTA with pH 5. The infection efficiency (percentage of GFP-positive cells) was determined using a cytofluorimeter (BD Accuri C6) and analysed using FlowJo software V10 (BD). To calculate the titre of the pseudovirus, expressed as TU/mL (transduced unit/mL), we considered the percentage of infected cells (GFP-positive cells), number of cells and the dilution of virus preparation, according to the following equation:Titre (TU/mL) = (N × P)/(V × D)where N = the cell number in each well used for infection on day 1; P = percentage of GFP-positive cells; V = the virus volume used for infection in each well, for which V (mL) = μL × 10^−3^; D = the dilution fold; and TU = the transduction unit.

We tested the kinetics of the pseudovirus at 48 h and 72 h, and we found that the transduction efficiency was higher at 72 h. Therefore, we optimised the assay under these conditions.

A standard lentivirus expressing the vesicular stomatitis virus (VSV) envelope plasmid (construct pMD2.G) was used as the control of transduction.

### 2.4. SARS-CoV-2 Pseudovirus Neutralisation Assay

Two assays were simultaneously developed, one using flow cytometry (FC) and a second using high-content screening imaging.

#### 2.4.1. FC Protocol

HEK293-ACE2 cells were seeded overnight in a 96-well culture plate (SAESTEDT, Ref#83.3924) at 3 × 10^5^ cells/well. Fivefold serial dilutions of serum or CP in duplicates were prepared in DMEM-2% FBS and mixed with an equal amount of pseudovirus preparation for a final starting dilution of 1:10, followed by incubation 1.5 h at 37 °C and 5% CO_2_. Afterwards, the supernatant from cell plates was replaced with 200 μL of virus-antibody mixtures, followed by 72 h incubation at 37 °C, 5% CO_2_ and 80% humidity. After 72 h, cells were harvested and resuspended in PBS (5 mM EDTA), and the percentage of transduced cells was analysed by FC to calculate the percentage of infection reduction and neutralisation titre, according to the following equation:Percentage of infection reduction = 100 × (1 − (GFP_diluted test sample_-GFP_no-virus control_/GFP_no-antibody control_-GFP_no-virus control_)).

Neutralisation titre was expressed as reciprocal to dilution, at which 50% of infection reduction is achieved.

Controls included sera from negative pre-COVID-19 donors (dilution 1:12.5), the virus alone and nontransduced cells.

#### 2.4.2. High-Content Imaging (HCI) Protocol

Huh7—hACE2 cells (4 × 10^3^ cells/well) were plated onto PerkinElmer 96-well plate and incubated at 37 °C, 5% CO_2_, overnight. Pseudovirus was mixed with fivefold serial dilutions of serum at a 1:1 ratio (100 µL/100 µL), incubated at 37 °C and +5% CO_2_ for 90 min and added to each well for spin infection, where pseudovirus-only wells served as positive controls and media-only wells and sera from negative pre-COVID-19 donors (dilution 1:12.5) as negative controls. After 48 h of incubation, plates were fixed with 4% PFA (paraformaldehyde) for 20 min at room temperature and washed twice with PBS 1x. Cells were treated with 4′,6-diamidino-2-phenylindole (DAPI, dilution 1:1000) and incubated for 30 min at 37 °C. Each plate was washed twice with PBS 1x. All plates were filled up with 150 μL of PBS/well. Digital images were acquired using the HCI system (PerkinElmer). The digital images were taken from 9 fields of each well at 20× magnification. The total number of cells and the number of transduced cells with GFP were analysed by using Columbus Image Data Storage and Analysis System (PerkinElmer) [23].

The GFP-positive cell percentages were evaluated using the following formula:Number of Positive Cells/Total Number of Cells × 100.

The percentage of neutralisation at different serum dilutions was calculated by the formula:(1 − (% of positive GFP cells of sample dilution − % positive GFP cells of negative control)/(% of positive GFP cells of positive control − % positive GFP cells of negative control) × 100.

The % neutralisation = 100 − % of transduction. NT_50_ titre was the serum dilution that reached 50% neutralisation using Excel nonlinear regression analysis (Excel Office 16) [27,28].

### 2.5. SARS-CoV-2 Spike Receptor-Binding Domain ELISA

Recombinant RBD was prepared according to a previous protocol [29]. Here, 96-well ELISA plates were coated overnight, at 4 °C, with 100 µL of 1 µg/mL purified recombinant RBD protein in PBS buffer [26]. The plates were then washed with 0.05% Tween 20 in PBS, blocked with 200 μL of 3% milk in PBS-Tween 0.05% and incubated at room temperature for 2 h. Each serum or plasma sample was tested in duplicate; twofold serial dilutions, from 1:50 to 1:25,600, were performed in 1% milk in PBS-Tween 0.05%; and 100 µL was added to the wells of each plate for 2 h incubation at room temperature. After 3 washes with PBS-Tween 0.05%, 100 µL horseradish peroxidase (HRP)-conjugated goat antihuman IgG γ-chain (Sigma-Aldrich, Product Number: A6029), in 1% milk in PBS-Tween 0.05% (1:5000) was added for 1 h at room temperature. The ELISA plates were washed 3 times with PBS containing 0.05% Tween 20. Subsequently, 50 μL of TMB substrate (Sigma-Aldrich, Product Number: T444) was added to each well. After 10 min incubation, the reaction was stopped by adding 50 μL of 1 M H_2_SO_4_ solution. Each well’s optical density (OD) was immediately measured by using a BIORAD iMark absorbance microplate reader at 450 nm wavelength. In total, 96 pre-COVID-19 sera were tested in duplicate at 1:50 dilution. The results of these samples were used to calculate the cut-off according to the following formula: OD Mean + 3 SD (standard deviation) [30].

### 2.6. Statistical Analysis

Statistical analyses were performed in R software (version 4.2.1, R Foundation for Statistical Computing, Vienna, Austria) [31]. We used the Q–Q plots and the Shapiro–Wilk test to assess the normality of the data distributions. The distribution of variables influenced the choice of parametric or nonparametric statistical tests.

To assess the associations between titres of serological tests, we divided the participants into groups of convalescent individuals and vaccinated individuals. The vaccinated participants’ group was further divided into individuals who had received only vaccination and individuals with both prior infection and vaccination. We evaluated the associations of titres obtained by ELISA, FC and HCI with Spearman’s (*ρ*) rank correlation coefficient. Additionally, we investigated the differences in titres across different tests and participants’ groups with the Wilcoxon rank sum test. The Wilcoxon effect-size metric determined the effect size. We used a false discovery rate (FDR) for multiple comparison corrections to address the multiple comparison problems.

The threshold for statistical significance was set at *p* < 0.05 in all cases. Excel nonlinear regression analysis (Excel Office 16) was used to calculate the antibody titres in all three methods: ELISA, FC and HCI. The levels of antibodies were plotted using GraphPad Prism (GraphPad Software, San Diego, CA, USA).

## 3. Results

### 3.1. Information on Samples

COVID-19 convalescent individuals, with a median age of 35 years (range from 20 to 59 years), were recruited during September–October 2020. All convalescent individuals initially showed symptoms via a computed tomography (CT) scan and were positive on SARS-CoV-2 nucleic acid testing.

Sera from 100 vaccinated people (24 men (17%) and 66 women (33%)) were collected in May 2021; of these, 26 individuals (5 men and 21 women) had prior COVID-19 vaccination. The age of immunised subjects ranged from 19 to 49, with a median of 22 years. In total, 96 additional healthy donors were recruited for this study.

A limitation of this study was that the detailed demographic information of the different groups could not be obtained.

### 3.2. Expression Level of ACE2 Receptor in the HEK293-ACE2 Cells

The expression level of the ACE2 receptor displayed on the membrane of HEK 293-ACE2 was verified through immunostaining and FC analysis. This check was carried out to ensure a good level of transduction in experiments with SARS-CoV-2 pseudotyped lentivirus. In this experiment, the negative control of expression was included by using wildtype HEK 293T cells incubated with the same antibodies. The results clearly show a positive fluorescence peak shift (Figure 1a) in HEK 293-ACE2 cells, incubated with both primary goat IgG anti-ACE2, and secondary FITC-labelled antibodies. In comparison, the same plot (A) shows two other negative overlapping peaks corresponding to the secondary antibody only (red peak) or to no antibody (blue peak). As expected for the control HEK 293T cells (Figure 1b), no significant peak shift is observed.

### 3.3. Production of SARS-CoV-2 Pseudotyped Lentivirus

GFP fluorescence was readily observed 72 h after transduction compared with nontransduced cells (Figure 2). To detect antibodies that recognise the spike protein in native conformation, a second-generation HIV-LV system was used. This system used the lentiviral transfer plasmid plVTHM encoding for the reporter gene (eGFP), the HIV packaging plasmid psPAX2 (encoding for gag-pol) and a plasmid carrying the sequence encoding for the structure glycoprotein spike, *pcDNA3*-SARS-CoV-2-Spike-D614GΔ19. The VSV lentivirus was used as a positive control of transduction because it can infect both HEK 293-ACE2 and HEK 293. In contrast, the SARS-CoV-2 lentivirus can infect only HEK 293 because it expresses the ACE2 receptor on the membrane.

### 3.4. SARS-CoV-2 and VSV Lentiviruses Titres Defined by FC

To determine the LV titre, serially diluted VSV and SARS-CoV-2 lentiviral preparations were incubated with 3 × 10^5^ HEK293/ACE2 cells, and the infection ratios were determined 72 h after incubation. Because only infected cells express the GFP protein, we can discriminate between transduced and nontransduced cells (Figure 3) by analysing the green fluorescence signal. In contrast, the percentage of GFP-positive infected cells (Figure 4) is related to the amount of virus produced.

The values showing the most linearity along the twofold dilution curve were chosen to calculate the titre. SARS-CoV-2 lentivirus preparation had a titre around ~3 × 10^5^ TU/mL (average measured on five independent experiments of viral preparation), and this titre was about one order of magnitude lower [27] than those achieved with VSV lentivirus, in which a titre of around ~6.69 × 10^6^ TU/mL was obtained.

### 3.5. SARS-CoV-2 Neutralising Antibodies after Natural Infection or Following Vaccination

Next, we set out to investigate the neutralising activity of antibodies elicited by the Sinopharm vaccine or by natural infection. To this end, we set up two SARS-CoV-2 pseudotyped lentivirus neutralisation assays, one using HCI microscopy and another FC.

In total, 296 participants were recruited in this study: 100 specimens of plasma from confirmed SARS-CoV-2-infected patients collected at 14 days postrecovery and whose PCR test was negative; 100 specimens from vaccinated individuals, taken 14 days after they had received their second dose of Sinopharm vaccine; and 96 pre-COVID-19 sera collected in 2018 by the National Transfusion Centre from the Republic of Moldova.

Samples were tested in duplicate at three dilutions (1:10, 1:50 and 1:250), which were mixed and preincubated for 2 h with an equal amount of SARS-CoV-2 pseudotyped lentivirus. After that, the mix was transferred to the target cells, which were incubated for 48 h (in HCI protocol) or 72 h (FC Protocol) to measure the neutralising antibody content of the sample.

Figure 5a shows representative images from HCI, and Figure 5b shows the FC results. In the upper two quadrants, the two control plots are shown, one for nontransduced cells (NT, without pseudovirus) and one for transduced cells in the presence of pre-COVID-19 sera, at a dilution of 1:10. In the FC analysis, the green fluorescence background, as determined in all experiments, for NT cells was around 0.2–0.5%. SARS-CoV-2 lentivirus with or without pre-COVID-19 sera (not shown) transduced around 28–40% of the cells [32]. Similar results were obtained on the HCI. The analysis showed in the negative control that 0.04–0.2% of the cells were positive, and in the positive control, the percentage of GFP-positive cells ranged from 19.80 to 42.01.

Thanks to both methodologies, it was possible to classify all the tested samples into three categories (Figure 5): (i) non-neutralising or weakly neutralising serum (sample #34CP); (ii) moderately neutralising serum (sample #43S); and (iii) strongly neutralising serum (sample #37CP).

We evaluated the level of NAbs through the reduction of GFP-positive n cells in three dilutions of the samples. A nonlinear regression analysis was used to determine half-maximal neutralisation titres (NT_50_).

For both conditions, vaccinated and convalescent, only 20% of the samples efficiently neutralised the pseudotyped SARS-CoV-2 at titres above 1:250. In comparison, for 50% of the sera, the response was weak or absent when determined by either FC or HCI (titres ranging from 1:10 to 1:50). Moreover, there is no significant difference between the vaccinated patients and the convalescent patients (see Figure 6).

NT_50_ in both studied groups, including methods applied for identification, are shown in Table 1. FC assay provides a median of 40.1 (range: 1–2182) for the level of NAbs compared with 27.6 (range: 1–1819) in convalescent patients.

The HCI protocol yielded similar results. The median titre of vaccinated individuals was 60.9 (range: 1–10,451), while for convalescent patients, the median titre was 24.6 (range: 1–11,051).

### 3.6. SARS-CoV-2 RBD-Specific Humoral Immunity in COVID-19-Recovered Subjects and in Vaccinated Individuals

The RBD-specific ELISA titres of serum in prepandemic healthy controls versus COVID-19-recovered and vaccinated subjects are shown in Figure 7. The OD threshold to calculate RBD IgG titres was 0.0819, and the IgG titres were defined as the reciprocal of the last dilution, at which the OD_450_ was above the threshold. Compared with prepandemic healthy controls, the immunised and COVID-19-recovered individuals showed potent and specific serologic activities towards RBD binding. At the same time, the Sinopharm-vaccinated individuals elicited higher anti-RBD SARS-CoV-2 IgG antibody levels compared with the convalescent patients, with median titres of 1742 versus 1239, respectively (Table 1).

Consistently, we also found that SARS-CoV-2 RBD-specific IgG antibodies were present in the sera of all vaccinated subjects, while there was one convalescent individual (60CP) who had undetectable titre values. The latter appears to be a healthy nonresponder who did not produce antibodies after recovering from COVID-19.

### 3.7. Correlations between ELISA, FC and HCI

SARS-CoV-2 spike RBDs are the main structural domains for inducing NAbs and play key roles in T-cell immune responses [33]. To define whether the neutralising antibody titres obtained by FC or HCI and RBD IgG antibody levels were correlated, we compared all three methods, and the results showed a similar correlation between the tests (Table 2).

The highest correlation coefficient was between ELISA and FC (ρ = 0.64), and the lowest was between ELISA and HCI (ρ = 0.51). We also observed similar correlation coefficients when we compared the methods with samples divided into convalescents and vaccinated. The relationship between titres and groups in ELISA RBD and FC is shown in Figure 8 and Figure 9. Even though some samples do not show detectable levels of NAbs, they still have binding antibodies. This indicates that a significant portion of the antibodies does not neutralise the virus, but there is a correlation between binding and neutralization. This is because there is a substantial population of antibodies that do have neutralising activity. Furthermore, the likelihood of detecting neutralisation increases with higher binding titres.

We compared one test for IgG RBD Ab and two for NAbs. When comparing the NT tests, the results showed a similar correlation between the tests as when comparing the serological tests with the NT tests. The highest correlation coefficient was in the convalescent group, between ELISA and FC (ρ = 0.68), and the lowest in the convalescent group, between ELISA and HCI (ρ = 0.46) (Figure 10a). The correlation coefficient (ρ = 0.55) was lower than the coefficient between ELISA and FC and higher than that between ELISA and HCI.

We divided the subjects into convalescent and vaccinated groups to define whether neutralising titres obtained by HCI and antispike RBD IgG levels were correlated. The correlation coefficient was ρ = 0.52 when comparing the overall titres in both studied groups (Figure 10a), but we noticed a higher correlation in the vaccinated group (ρ = 0.53) compared to the convalescent group (ρ = 0.45) (Figure 10b).

In this study, we also compared samples from convalescent patients and vaccinated individuals. Both groups of samples were tested with all three tests (Table 1). The results showed a statistically significant difference between the groups only in the HCI assay (*p* < 0.001). The median titre of the vaccinated individuals was 60.9 (range: 1–10,451) and was significantly higher than that in the convalescent patients, where the median titre was 24.6 (range: 1–11,051), *p* < 0.001. The result of our study is consistent with previous publications, which have shown that vaccinated individuals have higher antibody titres than convalescent patients do. The difference between the groups was observed only in HCI tests, not by FC, although this test also detects NAbs. Perhaps this could explain the moderate positive correlation (ρ = 0.55) between these two tests **(**Figure 11a). When separately analysed, the correlation coefficient between FC and HCI was (ρ = 0.58) in the vaccinated group, and (ρ = 0.51) in COVID-19-recovered individuals (Figure 11b; Table 2).

Additionally, we distinguished in the group of vaccinated individuals between participants with prior infection with SARS-CoV-2 and participants who were pathogen naïve when the vaccine was administered. We observed consistently higher titres in individuals with a higher number of immunological events (virus contraction, vaccination and both). However, statistical significance was demonstrated only by HCI titres between convalescent and vaccinated individuals (*p* < 0.05) and between convalescent and vaccinated individuals who were priorly exposed to the infection (*p* < 0.001). The comparison is presented in Figure 12.

## 4. Discussion

This work aimed to provide quantitative tools for sera analysis and to test a not-widely-used vaccine, at least in Western countries, and with limited information on its efficacy. BBIBP-CorV has been used in several countries, but less research has been conducted on it compared to Pfizer BioNTech’s BNT162b2, Moderna’s mRNA-1273, AstraZeneca’s AZD1222 and Pfizer’s BNT162b2 [34].

The Republic of Moldova’s immunisation process raised questions about vaccine efficacy and antibody levels, notably the Sinopharm vaccine. The Sinopharm vaccine partially protected against SARS-CoV-2 in Bahrain [35]. However, in a cohort study conducted in the United Arab Emirates, two-dose BBIBP-CorV vaccination efficacy against COVID-19 was 79.8% against hospitalisation and 92.2% against critical care admission compared to a nonvaccinated group [36]. Vokó Z et al. found that inactivated Sinopharm prevented COVID-19-related death, in countrywide observational research in Hungary [37].

In April 2020, the European Commission recommended experimental COVID-19 CP transfusions [38]. Several observational studies suggested that CP recipients benefited and that the donor’s antibody levels determined its efficiency [39].

Our work included 100 COVID-19-recovered individuals and 100 fully vaccinated recipients. All immunised human subjects generated IgG RBD antibody responses with median tiers of 1742. In contrast, among 100 CP subjects, one did not develop anti-RBD spike IgG antibodies, and the titres were lower (median: 1239) when compared to the vaccinated group. Our results bolster prior reports that have shown that the blood from donors who completed two doses of vaccines had higher RBD antibody levels than those of the convalescent group [40], but they assessed anti-RBD antibodies elicited by mRNA vaccines (Pfizer or Moderna) rather than the whole inactivated vaccine. A direct comparison of the antibody response to different vaccines showed that lower antibodies and neutralising activity were observed with the Sinopharm vaccine compared to other, more-widely-used vaccines [41].

Few of our participants (20%), in both studied groups, developed high-neutralisation titres to pseudotyped SARS-CoV-2 lentivirus. Most of them (50%) had moderate to low levels of NAbs, where titres ranged from 1:10 to 1:50. Similar results were reported by Omran EA et al. [42] in a study conducted in Egypt, demonstrating that only 67.4% of participants were positive for NAbs compared to 95.7% of subjects who tested positive for anti-S. These data suggested that while most COVID-19-recovered patients and Sinopharm-vaccinated subjects developed anti-RBD antibodies, only a small number can block pseudotyped SARS-CoV-2 lentivirus from attaching to the hACE2 receptor. Our results indicated that there was a significant, albeit moderate, correlation between the anti-RBD antibody level and the overall neutralisation activity against RBD-ACE2 binding (correlative analysis): ρ = 0.64, *p* < 0.001, when analysed by FC; and ρ = 0.52, *p* < 0.001, when analysed with the HCI protocol. In contrast, other studies have reported that anti-S and neutralising antibody titres had a strong correlation (ρ = 0.875–0.819, *p* < 0.001) [42,43]. Another study reported that NAbs were significantly correlated with the number of IgG RBD SARS-CoV-2 antibodies [44].

Early evidence suggested that several available antibody tests correlated poorly with neutralising titres [45]. In the current study, we developed an IgG ELISA with recombinant SARS-CoV-2 spike RBD and two BSL-2-safe assays with no infectious pseudovirus lacking the no-structural proteins. The pseudovirus system for screening NAbs and antivirals is a widely used method [46,47,48,49]. Most reports validate protocols based on ELISA and FC [28,32,50]. Because FC was time-consuming and required several steps in its analysis, we included a different technology, the HCI system. Following the HCI protocol, 15 samples can be simultaneously detected in a 96-well plate, while by FC cells from every well should be collected in an individual FC tube and independently analysed. High-throughput approaches are widely applied for drug discovery campaigns and have been recently implemented in neutralising assays [20,51,52]. We modified a 96-well HCI technique for SARS-CoV-2 [23]. Despite the strong heterogeneity of the antibody response, we found a general trend of increasing anti-RBD antibody levels with an increasing neutralising titre, but the results from the two neutralisation immunoassays showed different median NAb concentrations, confirming the need for standardisation to report data in a conventional format for comparison.

Even though we demonstrated that the Sinopharm vaccine induces higher anti-RBD antibody and neutralising antibody levels than natural immunity against SARS-CoV-2, future studies must include longitudinal titre levels over a broad time frame in order to gain a better understanding of the changes over time. This information helps guide policymakers to implement public health measures and works towards the early detection of the re-emergence of SARS-CoV-2. In this context, more studies in this field are required as new variants of the virus are detected. In addition to viral neutralisation, other hosts’ immune antiviral functions should be studied, such as complement activation, antibody-dependent cellular cytotoxicity and phagocytosis [53].

## Figures and Tables

**Figure 1 vaccines-11-00637-f001:**
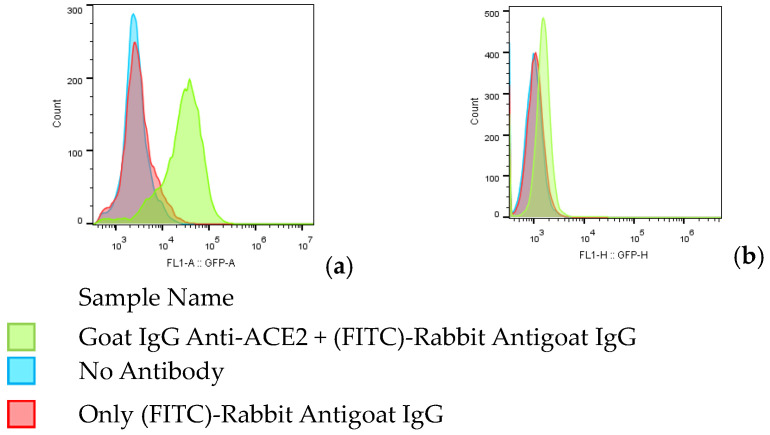
Expression levels of ACE2 receptor in HEK 293-ACE2 cells (**a**) and HEK 293T cells (**b**).

**Figure 2 vaccines-11-00637-f002:**
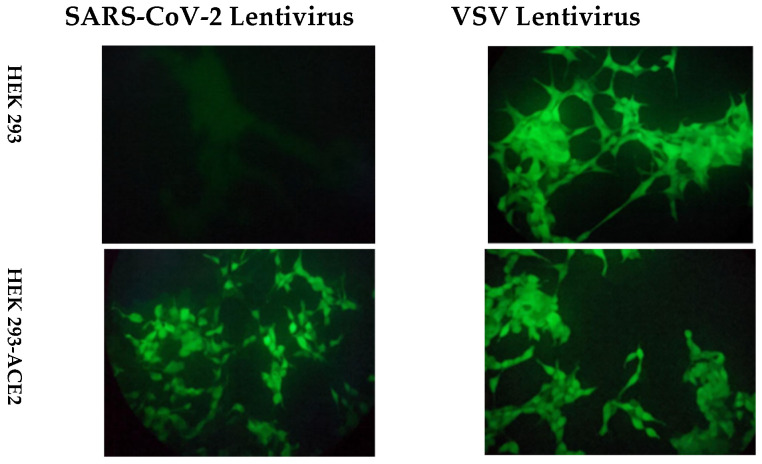
GFP expression following LV transduction, indicating the specificity of SARS-CoV-2 spike-expressing LV in binding HEK 293-ACE2 cells.

**Figure 3 vaccines-11-00637-f003:**
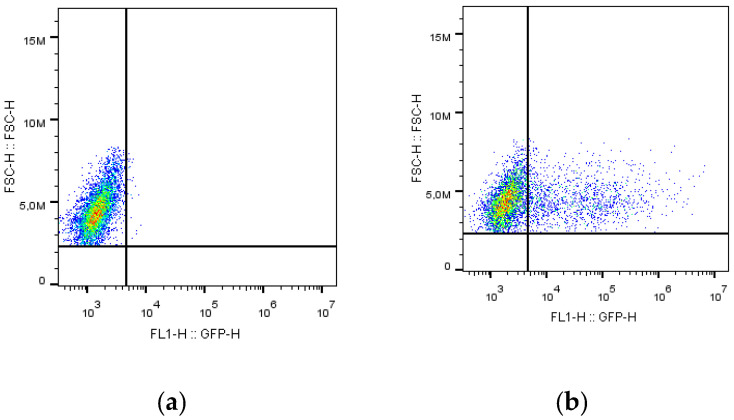
Green fluorescence analysed by FC of HEK 293-ACE2 cells under two conditions: nontransduced (**a**) and transduced (**b**) with SARS-CoV-2 lentivirus.

**Figure 4 vaccines-11-00637-f004:**
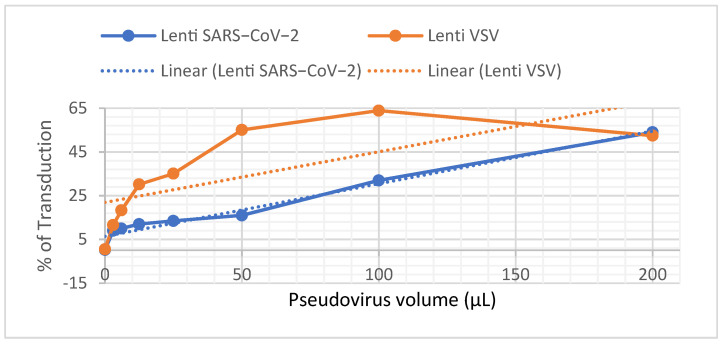
Titration of LV spike SARS-CoV-2 and LV VSV on HEK293-ACE2 cells.

**Figure 5 vaccines-11-00637-f005:**
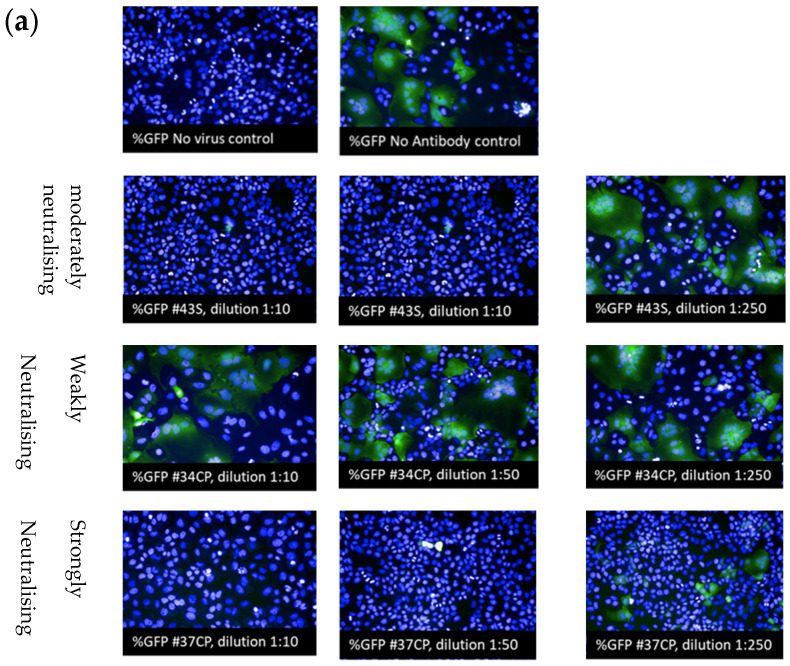
Detection of SARS-CoV-2 S-neutralising antibodies by high-content screening microscopy and FC. The first row shows images of positive controls: nontransduction or negative control (transduction with 200 µL of pseudoviruses without serum). The following rows show the results of the dose dilution response from three serum patient samples: (i) weakly neutralising serum (sample #34CP); (ii) moderately neutralising serum (sample #43S); (iii) strongly neutralising serum (sample #37CP). (**a**) Representative images from HCI showing the results obtained from the neutralisation assay. Nuclei are stained with DAPI (blue); positive transduction cells are detected from the GFP signal (green). One spot represents one cell, and GFP-positive spots correspond to infected cells. (**b**) Representative images from FC showing the results obtained from the neutralisation assay. Numbers in the FC plot indicate the percentage of GFP+ cells in the respective quadrants. Data are from a single experiment representative of three independent experiments.

**Figure 6 vaccines-11-00637-f006:**
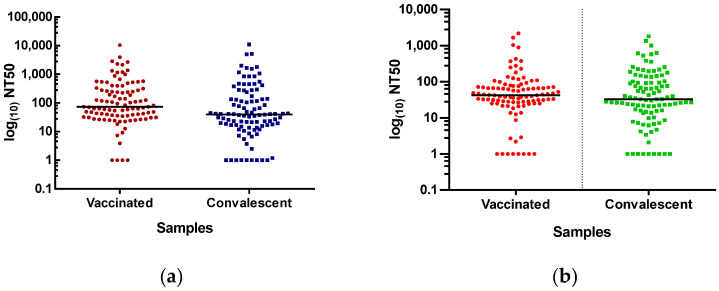
SARS-CoV-2-neutralising antibody titres obtained in the Sinopharm-vaccinated cohort and COVID-19-recovered patients. Scatter plot shows the results obtained (**a**) by HCI protocol and (**b**) by FC. The black line corresponds to the median. For visual purposes, 25 extreme values are not shown in the plot but are kept in the calculations. Healthy donor sera were tested at a single 1:12.5 dilution, and all showed an NT_50_ < 0.94.

**Figure 7 vaccines-11-00637-f007:**
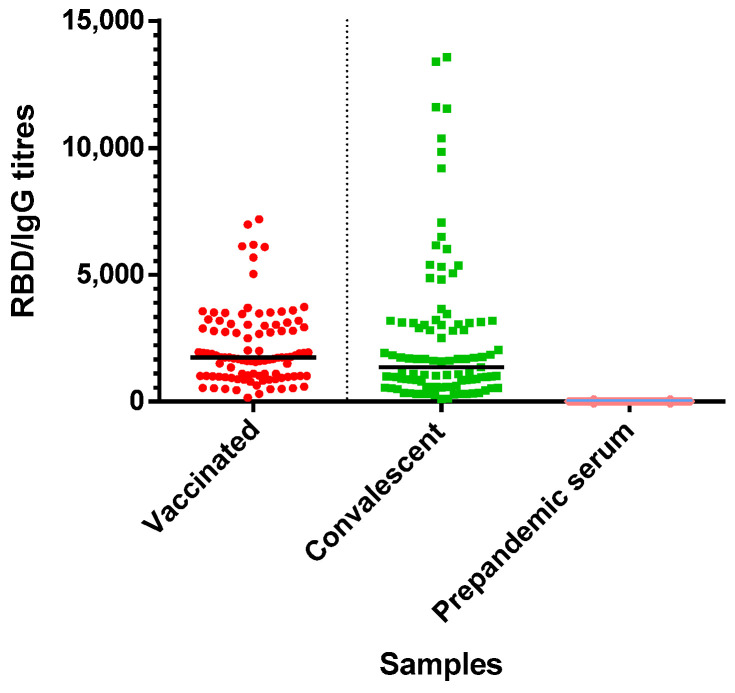
Anti-SARS-CoV-2 RBD IgG antibody titres in 100 Sinopharm-vaccinated recipients (vaccinated), 100 COVID-19-recovered individuals (convalescent) and 96 seronegative subjects (prepandemic serum). The horizontal black line corresponds to the median.

**Figure 8 vaccines-11-00637-f008:**
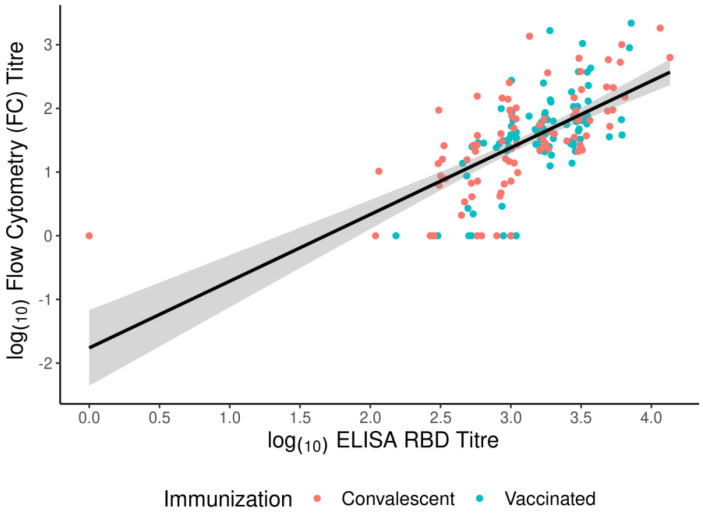
Log10 transformed values of titres obtained from ELISA RBD and FC. Different colour points indicate convalescent (red) and vaccinated (blue) participants. The best-fit linear regression line with a corresponding confidence interval is presented as a black line with a grey band.

**Figure 9 vaccines-11-00637-f009:**
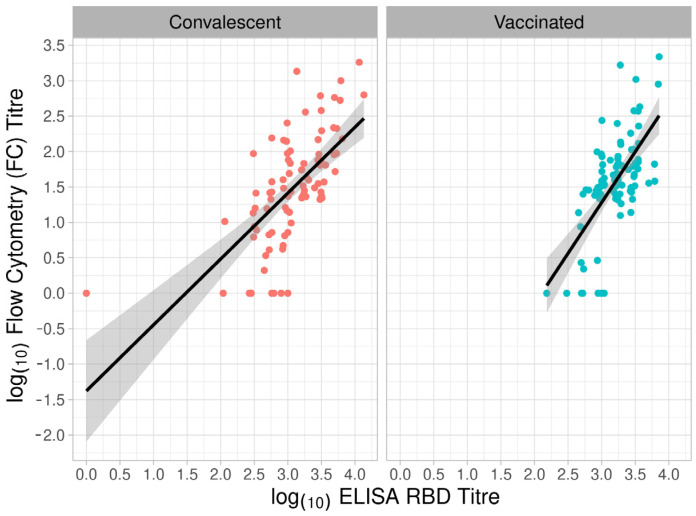
Log10 transformed values of titres obtained from ELISA RBD and FC. Different colour points indicate convalescent (red) and vaccinated (blue) participants. The best-fit linear regression line with the corresponding confidence interval is presented as a black line with a grey band.

**Figure 10 vaccines-11-00637-f010:**
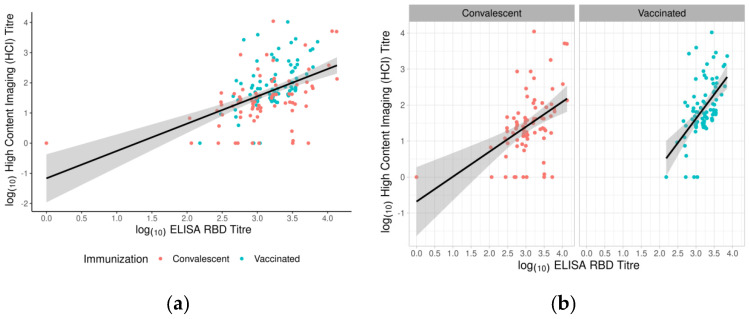
Anti-RBD IgG antibodies in human sera and their correlation with neutralising titre. (**a**) Log10 transformed values of titres obtained from ELISA RBD and NT_50_ determined by HCI. (**b**) Correlation between anti-RBD IgG and neutralising antibody levels in convalescent subjects and vaccinated subjects. Different colour points indicate convalescent (red) and vaccinated (blue) participants. The best-fit linear regression line with the corresponding confidence interval is presented as a black line with a grey band.

**Figure 11 vaccines-11-00637-f011:**
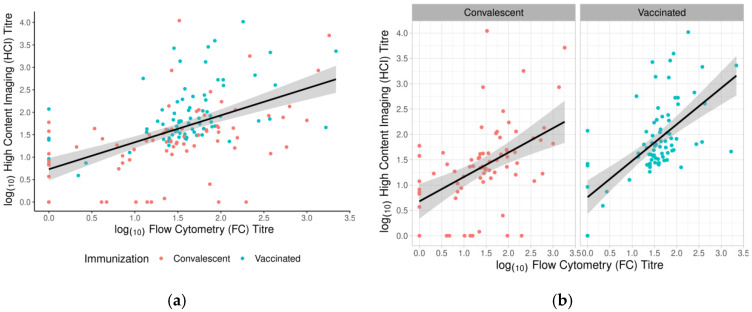
Correlation of neutralising antibody titres defined between FC and HCI for each studied group. (**a**) Log10 transformed values of NT_50_ determined in both studied groups. (**b**) Correlation of neutralising antibody levels between two neutralisation assays, in convalescent subjects and vaccinated subjects, separately. Different colour points indicate convalescent (red) and vaccinated (blue) participants. The best-fit linear regression line with the corresponding confidence interval is presented as a black line with a grey band.

**Figure 12 vaccines-11-00637-f012:**
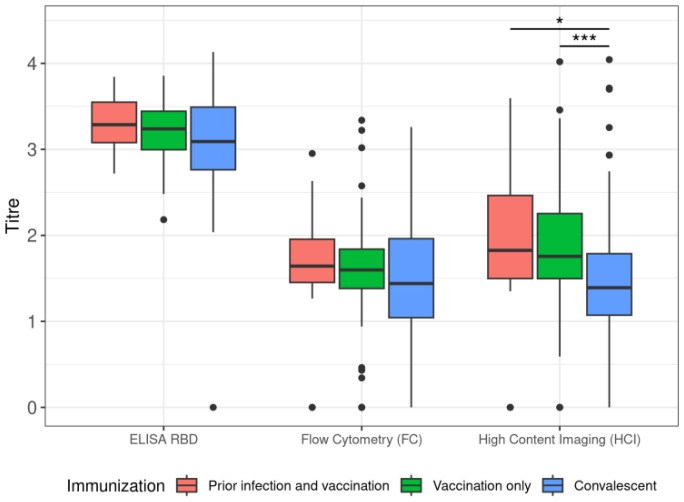
A comparison of titre distribution between the used methods and different subgroups of our participants. * *p* < 0.05; *** *p* < 0.001.

**Table 1 vaccines-11-00637-t001:** Median and range of quantitative antibody titres between methods and studied groups/subgroups.

	Overall Titre	Titre in Convalescent	Titre in Vaccinated	Titre in Vaccinated (+ Prior Infection)	Titre in Vaccinated (Naïve)
ELISA IgG RBD; median (range)	1678 (1–13,565)	1239 (1–13,565)	1742 (152–7184)	1936 (524–6978)	1731 (152–7184)
FC; median (range)	35.8 (1–2182)	27.6 (1–1819)	40.1 (1–2182)	43.9 (1–896)	39.6 (1–2182)
HCI; median (range)	41.3 (1–11,051)	24.6 (1–11,051)	60.9 (1–10,451)	67 (1–3940)	57 (1–10,451)

**Table 2 vaccines-11-00637-t002:** Comparison of SARS-CoV-2-specific antibody responses versus neutralisation titres, through various methods.

Group	Method_1	Method_2	Correlation_Coefficient (Spearman)	*p*_val
All	ELISA_RBD	FC	0.64	<0.001
Convalescent	ELISA_RBD	FC	0.68	<0.001
Vaccinated	ELISA_RBD	FC	0.58	<0.001
All	HCI	ELISA_RBD	0.52	<0.001
Convalescent	HCI	ELISA_RBD	0.45	<0.001
Vaccinated	HCI	ELISA_RBD	0.53	<0.001
All	FC	HCI	0.55	<0.001
Convalescent	FC	HCI	0.51	<0.001
Vaccinated	FC	HCI	0.58	<0.001

## Data Availability

Not applicable.

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
