# Peer review of "Characterisation of the Antibody Response in Sinopharm (BBIBP-CorV) Recipients and COVID-19 Convalescent Sera from the Republic of Moldova"

_vaccines, 2023, doi:10.3390/vaccines11030637_

Round 1
Reviewer 1 Report
This manuscript presents a study that set up the methodology for Ab detection against RBD in ELISA and nAb against SARS-CoV-2 pseudovirus. The authors then reported their findings using the methods to measure these Ab in sera from COVID-19 convalescent patients and subjects receiving Sinopharm inactivated SARS-CoV-2 vaccine in Moldova.
Major Points:
1. There is no detailed demographic information of the subjects, and I do not know if the different groups in this study can be compared without bias. Authors may need to be cautious in comparison or write a passage on the limitations of this study regarding this.
2. What is the spike sequence in the pseudovirus? Wuhan-hu-1 or other VOC?
3. The convalescent patients were infected by what strain of the virus? This information is important since if the patients were infected in e.g., a delta wave, their sera may not show much neutralization of the Wuhan-Hu-1 spike-pseudotyped lentivirus.
Minor points:
1. Fig 4. Unit of X-axis?
2. Line 320 and 322, "column" should be "row"
3. Fig. 6, Y axis title should be 50% ID or use the NT50 in the text. And what is the nAb titer for healthy donors?
4. Fig. 10, titles of the axis is too tiny, unreadable
5. L 414 and 416, Fig 10 should be Fig.11
6. L424, Table 1 not Table 2, and also the text mixed up the convalescent and vaccinated nAb values which is not consistent with the Table.
Reviewer 2 Report
This is a report assessing anti-SARS-CoV-2 antibody levels after Sinopharm vaccination and in COVI19 convalescent sera in a group of individuals from Moldova. Contrary to the authors’ conclusion, there does not seem to be much difference between the two groups nor in the titers observed which remain moderate if different from pre-pandemic sera. This study merely shows that the three assays developed by the authors yield interpretable and correlated data. Unfortunately, there is no comparison with i)patients vaccinated with EU authorized vaccines, although such patients exist in Moldova according to the authors’ information in the introduction and ii)assessment of the same samples with classical commercial tests used in many reports. It is therefore difficult to draw definitive conclusions as to the value of the Sinopharm vaccine and its comparability with literature data.
The introduction should focus on the topic considered and remove all other considerations of the management of the pandemic in Moldova.
How do the authors explain the long incubation times for flow cytometry?
How were cell percentages evaluated in the HCI assay?
Figure 5 cannot be readily interpreted in relation to the text since there is no indication of the patient # mentioned in the text.
On which basis do the author consider their data as a solid basis for future testing (lines 349 on)?
The legend of Figure 7 should better match the logic of its presentation. Moreover on which basis do the authors assert that titers were higher in vaccinated patients and what is the significance of this putative difference not apparent on the graph?
There is a misinterpretation in the comparison of data lines 426 on.
The discussion, similarly to the introduction, is far too long and out of the focus of this specific study.
Minor
Abbreviations should be defined on first instance and solely used afterwards.
The source of reagents (manufacturer, city and country or US state) should be given extensively on the first occurrence then only by manufacturer.
There are a few typos and English language misuse.
The reference list is heterogeneous, months and days of publication should be removed, titles harmonized with low case except for the first word and abbreviations, and the same style applied to all.
Reviewer 3 Report
This clinical virology paper reports a COVID-19 RBD ELISA and 2 neutralization assays to measure antibodies in COVID-19 convalescent sera and Sinopharm vaccine sera administered to students and staff at a University in Moldova. They find positive correlations between binding and neutralizing antibodies measured (using a GFP reporter assaying two formats), especially for convalescent sera and higher neutralization titers in vaccine sera.
Overall, the paper is technically sound and quite interesting.
Some comments and suggestions are given below:
Line 50: Is it Sinopharm or Sinovac? Sinopharm is the company and Sinovac the vaccine. Text needs editing here for clarity.
Fig. 1 a and b could be aligned better. This figure should not take up too much space and could be supplemental. The same goes for Figure 2: the figure concerns reagents and could be concise to not occupy too much space, or be put as a supplemental figure.
Figure 4: what is the x axis units?
Fig 6: should be fixed so both charts are sharply reproduced (part A is fuzz). Also, there are two bars on the left chart- what are they? One pair have color matching the symbols and the other pair are black. The black bars on the right chart do not appear to be half maximum. The position of the black bars is also an issue in Fig. 7. Does the y axis look better in log scale?, a lot of data is accumulated at low titers which is hard to evaluate. Vaccine sera appear to have titers >100 to RBD (Fig. 9 using log scale x axis). It is claimed vaccine sera median is higher than CP, but the black bars in the figure don’t jive with that assertion.
Font sizes and labels on the figures could be a bit larger. They are hard to see in some cases.
Is it surprising that many vaccine sera neutralization titers were quite low? It is expected that infection convalescence has a range of titers depending on disease severity, but variation for vaccinees is surprising. What adjuvant and dose is used? How does this compare to other vaccines like Moderna and Pfizer based on literature comparisons of covalescent sera neutralization versus neutralizing titers to these other vaccines? Samples are lacking, but it would be of interest to check the literature and discuss this point.
In analysis fig 8, 9, it could be mentioned that there are some samples with undetectable nAbs that still have binding antibodies, showing that a large proportion don’t neutralize, except that binding and neutralization correlate, because a sizeable population of antibodies does neutralize and the higher the binding titer, the more likely that neutralization is detected.
Round 2
Reviewer 2 Report
The authors have improved the manuscript according to the suggestions provided. Some minor points however still require attention.
Please use SI units, i.e. L not l mL not ml, µL not µl, a.s.o. (the idea is to assess L and avoid confusion with the letter I in caps.)
Use FC after the first definition of the acronym that should occur line 127, not in the title of 2.4.1.
Lines 35 and following, the formulation would be better as: “In the Republic of Moldova, the first 35 case of SARS-CoV-2 infection was registered on March 7, 2020 [3] and, as of November 27, 2022, 595,073 COVID-19 cases (15,955 per 100,000 inhabitants) had been registered, causing 11,918 deaths (319,6 deaths/100,000 inhabitants).
Line 89: “The vaccine contains 0.225 mg of” (omit ‘the’ before the dosage)
Lines 136 and following: this is where I wondered about the length of incubations for flow cytometry. This remark has not been answered.
Line 166: this should be moved up to 2.3
Line 214: “..normality of data distribution” (omit “the” before data) and the sentence could go on as “..influenced the choice of parametric…”
Lines 353 and following. Please shorten the legend and avoid the contradictory statements about the comparison of vaccinated and convalescent patients.
Table 2 spell ELISA not elisa and check the use of italics
The discussion has been reasonably shortened and focused
References are still not homogeneous (pages missing, update internet consultations)
Author Response
The authors have improved the manuscript according to the suggestions provided. Some minor points however still require attention.
I thank the reviewer for the positive and constructive comments and revised the manuscript accordingly.
Please use SI units, i.e. L not l mL not ml, µL not µl, a.s.o. (the idea is to assess L and avoid confusion with the letter I in caps.)
We have amended the SI units accordingly, we now use “mL” and “µL”
Use FC after the first definition of the acronym that should occur line 127, not in the title of 2.4.1.
We have made corrections as per referee suggestion. Now We use FC acronym after the 1st definition
Lines 35 and following, the formulation would be better as: “In the Republic of Moldova, the first 35 case of SARS-CoV-2 infection was registered on March 7, 2020 [3] and, as of November 27, 2022, 595,073 COVID-19 cases (15,955 per 100,000 inhabitants) had been registered, causing 11,918 deaths (319,6 deaths/100,000 inhabitants).
We corrected the formulation as per referee suggestion (Lines35-38).
Line 89: “The vaccine contains 0.225 mg of” (omit ‘the’ before the dosage)
Thank you for the comment, now the text is corrected
Lines 136 and following: this is where I wondered about the length of incubations for flow cytometry. This remark has not been answered.
Thank you for the comment. We remark was amended (lines143-144)
Line 166: this should be moved up to 2.3
We have followed your suggestion and moved the text to 2.3 (lines 143-144)
Line 214: “..normality of data distribution” (omit “the” before data) and the sentence could go on as “..influenced the choice of parametric…”
Thanks for the comment, this was corrected.
Lines 353 and following. Please shorten the legend and avoid the contradictory statements about the comparison of vaccinated and convalescent patients.
The legend was shortened and the text clarified. (lines 355-375)
Table 2 spell ELISA not elisa and check the use of italics
The table was adjusted according to the recomendations
The discussion has been reasonably shortened and focused
Thanks for positive feedback
References are still not homogeneous (pages missing, update internet consultations)
Thank you for this important comment. We now have homogenized that references, added pages and internet consultations dates.